# Using touchscreen equipped operant chambers to study animal cognition. Benefits, limitations, and advice

**Benjamin M. Seitz**[1]*, **Kelsey McCune**[2], **Maggie MacPherson**[2], **Luisa Bergeron**[2], **Aaron P. Blaisdell**[1], **Corina J. Logan**[3]*

1 University of California, Los Angeles, Los Angeles, California, United States of America, 2 University of California, Santa Barbara, Santa Barbara, California, United States of America, 3 Max Planck Institute for Evolutionary Anthropology, Leipzig, Germany

* bseitz1@ucla.edu (BMS); corina_logan@eva.mpg.de (CJL)

**Data Availability Statement:** The detailed training protocol is contained within the Supporting Information files, and a video summary of the training is available on YouTube (https://youtu.be/

## Abstract

Operant chambers are small enclosures used to test animal behavior and cognition. While traditionally reliant on simple technologies for presenting stimuli (e.g., lights and sounds) and recording responses made to basic manipulanda (e.g., levers and buttons), an increasing number of researchers are beginning to use Touchscreen-equipped Operant Chambers (TOCs). These TOCs have obvious advantages, namely by allowing researchers to present a near infinite number of visual stimuli as well as increased flexibility in the types of responses that can be made and recorded. We trained wild-caught adult and juvenile great-tailed grackles (*Quiscalus mexicanus*) to complete experiments using a TOC. We learned much from these efforts, and outline the advantages and disadvantages of our protocols. Our training data are summarized to quantify the variables that might influence participation and success, and we discuss important modifications to facilitate animal engagement and participation in various tasks. Finally, we provide a "training guide" for creating experiments using PsychoPy, a free and open-source software that was incredibly useful during these endeavors. This article, therefore, should serve as a resource to those interested in switching to or maintaining a TOC, or who similarly wish to use a TOC to test the cognitive abilities of non-model species or wild-caught individuals.

## Introduction

A number of scientific disciplines including behavioral neuroscience, experimental psychology, ethology, and ecology, aim to understand the cognitive abilities of animals. Given the so-called *black box paradox* of studying cognition, where behavioral measures are used to infer cognitive abilities, a number of technologies have been developed that measure behavioral responses to a variety of stimulus inputs. Perhaps the most influential technology developed to study animal cognition is the "operant chamber" (also referred to as the *Skinner Box*). While many variations exist, an operant chamber (OC) typically consists of a small enclosure with one or several manipulanda (e.g. levers, buttons, chains) and responses made to them are

NeqMb7jaDmc). The training data is available on the Knowledge Network for Biocomplexity data repository (doi:10.5063/F1J964SR).

**Funding:** This research is funded by the Department of Human Behavior, Ecology and Culture at the Max Planck Institute for Evolutionary Anthropology (https://www.eva.mpg.de/index.html) (2017-current), and by a Leverhulme Early Career Research Fellowship (https://www.leverhulme.ac.uk/early-career-fellowships) to CJL (2017-2018). The funders had no role in study design, data collection and analysis, decision to publish, or preparation of the manuscript. Benjamin Seitz is supported by National Science Foundation grant DGE-1650604 and Aaron P. Blaisdell is supported by National Science Foundation research grant BCS-1844144.

**Competing interests:** The authors have declared that no competing interests exist.

recorded by a computer. Critical to the OC is also the ability to present a variety of stimuli (e.g., auditory and visual cues) as well as biologically significant outcomes (e.g., food or shock) that are often contingent on the animal's interactions with the manipulanda. However, these technologies have been primarily used with captive-reared model species, thus limiting our ability to generalize results to naturalistic conditions (due to the intentional or unintentional artificial selection that is often involved [1]) and make cross-species generalizations about cognitive abilities.

As technologies advance, many researchers have transitioned to using touchscreen-equipped operant chambers (TOCs) because they confer a number of advantages (see below). TOCs are now commonplace in studies utilizing more traditional laboratory animals such as rodents, primates, and pigeons. TOCs have also been used to study non-model species such as bears [2, 3], dogs [4], and tortoises [5] in captivity. However, rarely has the use of this method been described in wild-caught individuals from non-model species (but see [6, 7]). There is a paucity of information about how to train such individuals to use a TOC, potentially because OCs and TOCs were tailored to the behavior and characteristics of model species, which could explain the difficulty in training non-model species and the increased variability in training success and performance. This distinction is relevant to the ability to generalize TOC procedures across groups because captive-reared animals often have different ontogenetic experiences that affect their motivation to interact with, and performance on, behavioral tasks compared to wild conspecifics [8, 9]. We embarked on an investigation using TOCs in wild-caught adult great-tailed grackles (*Quiscalus mexicanus*; hereafter grackles) that were temporarily held in aviaries. Extensive modifications to procedures used with laboratory pigeons were required to train grackles to use such an apparatus. Our aim here is to briefly describe the advantages and disadvantages of TOCs for animal cognitive testing, report data summarizing our efforts to train several grackles, and finally, to provide a "training guide" to facilitate the use of TOCs in non-model species while using open-source software.

## Advantages and disadvantages of touchscreen-equipped operant chambers (TOCs)

There are a number of advantages associated with using TOCs compared to traditional OCs or open field tasks. Notably, and as reviewed by others (e.g., [10–12]), TOCs allow for the presentation of virtually any combination of visual and/or auditory stimuli. Whereas OCs typically have a limited number of spatially-fixed response options (e.g., lever pressing or chain pulling), TOCs afford a much wider range of possible response locations [11]. This is beneficial not only when testing the same animal on multiple tasks but also for adjusting potential side biases and other response biases which are commonplace in operant conditioning procedures [10]. TOCs record not only that a response has been made, but also, precisely when in the trial and where on the screen the response was made, unlike traditional manipulanda found in OCs [13]. As such, TOCs are particularly useful for investigations of spatial learning and memory processes [14, 15].

TOCs allow for experimental procedures (e.g., protocols, code, methods) to be more easily shared between researchers [16]. The ease of sharing programs and data sets among research groups around the world, and the flexibility in stimuli and testing paradigms should facilitate the inclusion of novel species trained with TOCs as well as cross-species comparisons. A number of species have already been successfully trained using TOCs, which points to the flexible and intuitive nature of touchscreen tasks [10, 17]. In fact, there is evidence that touchscreens can be more effective in training animals on discrimination-based tasks than traditional OC methods, presumably because touchscreens allow the animal to interact directly with the

stimulus to obtain some immediate outcome (i.e., touch the stimulus to make a choice) instead of responding to a lever that is spatially separated from the stimulus [18, 19]. Additionally, progress on TOC tasks can be remotely viewed and controlled by an experimenter outside the periphery of the subject, so there is more information on performance during each session. Finally, because humans and non-humans can be given nearly identical behavioral tasks in terms of the stimuli presented and available responses (but not necessarily reinforcement received), TOCs can be powerful tools for translational research especially for studying neuro-psychiatric phenomena [20].

The most obvious barrier to adopting TOCs is that they require researchers to possess some level of programming skills. This can be difficult for researchers whose backgrounds are far removed from computer science. Some researchers have circumvented this issue by hiring programmers to build their programs, but this can be a costly and inefficient solution. More-over, the existence of multiple different programming languages, some of which are not free to use (e.g., E-Prime or MATLAB), makes it very difficult for experimenters to share experimen-tal paradigms with each other. This reduces the possibility of collaboration and replication. Another challenge of TOCs are accidental contacts made to the touchscreen. This often occurs when the bird's breast or tail makes contact with the screen and can be a potentially serious issue when unintentional actions lead to reward or punishment that affect learning processes and behavior [21]. Fortunately, programs can be formatted such that only responses made to specific stimuli (and not the entire screen) are recorded and/or result in some specific out-come. That said, accidental contact to stimuli does occasionally occur, which suggests that cou-pling this technology with a video recording device can be particularly helpful. Similarly, at least with avian species in our specific setup (see details below), not all pecks to the TOC are registered by the infrared system. This has less impact during the later stages of training, because birds will often make multiple pecks to the same stimulus, but it can be a challenge during initial training. Further, when a response is accidentally made or missed, there are lim-ited ways for correcting for this in real time (but see below for our suggested technique of cou-pling the experimental program with a livestream video call to remote control the operating computer). Researchers interested in building their own TOCs may wish to consult engineers to ensure their touchscreen equipment is sensitive to the types of responses made by their respective species of study.

## Lessons from teaching wild-caught animals to use touchscreens

From 2018 to 2020, we conducted an investigation of the behavior of wild-caught grackles [22–24] using a number of different experimental procedures, many of which involved using a TOC. Several unexpected hurdles in training grackles to use the TOC were encountered, which could be due to species differences and/or because they were wild-caught rather than captive-bred. Consequently, we gained valuable insights that are likely useful for other researchers interested in a similar approach. These insights cluster around two major themes: 1. Managing challenges that arise from training wild-caught adult birds to operate a TOC com-pared to traditional laboratory animals like pigeons; and 2. The benefit of using PsychoPy soft-ware to create behavioral tasks for TOCs. We discuss our process of navigating these themes below, use our TOC training data to posit relationships with training performance and the number of prior non-TOC experiments and TOC experiments completed, and finally detail our process for using PsychoPy software to design behavioral tasks on TOCs.

TOC testing opened up new avenues for the types of comparative cognition experiments we were able to conduct with temporarily-held wild birds, but required several modifications to allow the successful testing of grackles. The detailed grackle TOC training protocol can be

found here: https://tinyurl.com/yxurnle5, a video summary of the training is at https://youtu.be/NeqMb7jaDmc, and the training data are online in [25]. By incorporating TOC tests into the test battery, individuals were able to be tested with less interference from human experimenters (automation makes testing faster; see [23, 26] for discussion on benefits of free operant techniques), tests could be conducted that were difficult to implement without a TOC (e.g., Go/No-Go and causal cognition experiments), and the ecological relevance of previous TOC experiments could be validated where individuals from model species were used that have been captive for generations and thus might lack the behavioral responses that one would see in wild individuals that are subject to natural selection [27–29]. Additionally, if the apparatus and training can be effectively modified, it facilitates bringing the TOC to the field, which could remove the additional time and financial costs of temporary captivity for research programs that focus on wild individuals (e.g., for wild vs. captive reversal learning using operant boxes, see [30], for examples of automated feeders used in the wild, see [31, 32]).

## Methods

### Ethics

The research on wild-caught grackles was possible under the IACUC no. 17-1594R from Arizona State University, and through CL's US Fish and Wildlife Service Scientific Collecting permit (number MB76700A-0,1,2), Bird Banding Permit from USGS (number 23872), and a Scientific Collecting permit from the Arizona Game and Fish Department (SP594338 [2017], SP606267 [2018], SP639866 [2019], and SP402153 [2020]).

### Subjects

The 12 adult grackles (Fajita, Mole, Tomatillo, Habanero, Chalupa, Tapa, Adobo, Diablo, Burrito, Marisco, Queso, and Yuca) and 2 juvenile grackles (Taco and Chilaquile; see Table 1) that underwent touchscreen training were caught in the wild in Tempe, Arizona, USA, from September 2018 through November 2019 and temporarily brought into outdoor aviaries for

**Table 1. Summary of all of the experiments the grackles who experienced TOC training participated in.**

| Bird | Sex | Non-TOC: Reversal learning tubes | Non-TOC: assays exploration boldness | Non-TOC: Multiaccess box | Non-TOC: Detour | Non-TOC: demonstrator training | TOC: Reversal | TOC: Go/ no go | TOC: Causal cognition |
|---|---|---|---|---|---|---|---|---|---|
| Fajita | F | 1 | - | 2 | - | - | - | - | - |
| Tomatillo | M | 1 | 2 | 3 | 4 | - | - | - | - |
| Queso | M | 2 | 3 | 4 | 1 | - | 5 | 6 | - |
| Tapa | F | 2 | 3 | 4 | 1 | - | 5 | - | - |
| Yuca | F | 1 | 2 | 3 | 4 | 5 | - | 6 | 7 |
| Marisco | M | 1 | 2 | 3 | - | 4 | - | - | - |
| Chalupa | F | 1 | 2 | 3 | 4 | - | - | - | - |
| Mole | M | 1 | 2 | 3 | 4 | - | 5 | 6 | - |
| Habanero | M | 1 | 2 | 3 | 4 | - | 5 | - | - |
| Diablo | M | 1 | 2 | 3 | 4 | 5 | - | 6 | 7 |
| Burrito | M | 1 | 2 | 3 | 4 | 5 | - | 6 | 7 |
| Adobo | M | 2 | 3 | 4 | 1 | 5 | - | 6 | 7 |
| Chilaquile | JM | 1 | 2 | 3 | 4 | - | - | 5 | 6 |
| Taco | JM | 1 | 2 | 4 | 5 | 3 | - | - | - |

The number indicates the order of the experiments for that bird, while a "-" indicates that experiment was not offered to the bird. Sex categories: F = female, M = male, J = juvenile.

behavioral testing before being released back to the wild. Each bird was measured, then color-marked with leg bands in unique combinations for identification, and blood samples were drawn prior to being placed in the aviary as part of other research (see [22–24]). Grackles were individually housed in an aviary (each 2.44 m long by 1.22 m wide by 2.13 m tall) at Arizona State University for a maximum of six months. All subjects had *ad libitum* access to water at all times and were fed Mazuri Small Bird maintenance diet *ad libitum* during non-testing hours (minimum 20 h per day), and other, more preferred food items (e.g., goldfish crackers) during testing. Individuals were given a minimum of three days to habituate to the aviaries and then their test battery usually began on the fourth day. Birds were usually tested six days per week, therefore if their fourth day in the aviaries occurred on a day off, then they began testing on the fifth day instead. After engaging in three to four experiments (experiment order was pseudorandomized such that some individuals received TOC training earlier than others), grackles were then habituated to and trained to use the TOC, and subsequently participated in one to three TOC experiments. Unless habituation was occurring, the TOC was wheeled into the aviary, a training session was conducted, and then it was wheeled out of the aviary again. One bird, Queso, had a medical procedure (he had a lump on his head when he was captured, which the Arizona State University veterinarians drained) in the middle of white square training, which extended his training period because we waited for him to recover and then to become motivated to participate again. Some birds did not pass training because they had to be released back to the wild (Fajita, Tomatillo, Chalupa, and Taco).

### Test battery

Before beginning TOC training to prepare the birds for the TOC experiments (go/no go: see [24] for details; causal cognition: see [22]; reversal learning: see [23] these individuals experienced non-TOC experiments). Non-TOC experiments included a color tube reversal learning experiment and a puzzle box experiment (see [23] for details), a detour experiment (see [24]), exploration and boldness assays (see [33]), and demonstrator training for a social learning experiment ([see 34]) and see Table 1 for a complete summary.

### The touchscreen-equipped operant chamber

Our TOC consisted of a color LCD monitor (NEC MultiSync LCD1550M) measuring 23.2 cm x 30.5 cm that sat behind an infrared touchscreen (Carroll Touch, Elotouch Systems, Fremont, CA). A food hopper (a container filled with food) was located below the monitor with an access hole situated parallel to the floor. The hopper was raised and lowered using a robotic arm (Pololu Maestro 12 Channel USB Servo, Pololu Robotics, Las Vegas, NV) and, when in the raised position, the food inside the hopper was accessible. While a TOC set-up is typically enclosed by an opaque chamber, this was removed for the grackles to allow the birds to engage with the apparatus at will and thus not rely on researchers actively placing the bird in the OC for participation (see Fig 1 for a picture of our setup during hopper training). One piece of advice we offer is installing software (e.g., Teamviewer or Microsoft Remote Desktop) that allows one to remotely control the computer being used to operate the TOC. This provides a solution for the issue of a response not being detected by the touchscreen because the experimenter can use a different computer to register the response made by the animal on the TOC —either by observing from a distance or over live video feed. For example, by simply installing the TOC with a webcam and initiating a Skype (www.skype.com) call with the remotely-controlled computer, allows the experimenter to observe the bird's behavior in real time and intervene if necessary. For a detailed description of the grackle training procedures including a discussion of techniques and programs that did/did not work, see S1 File.

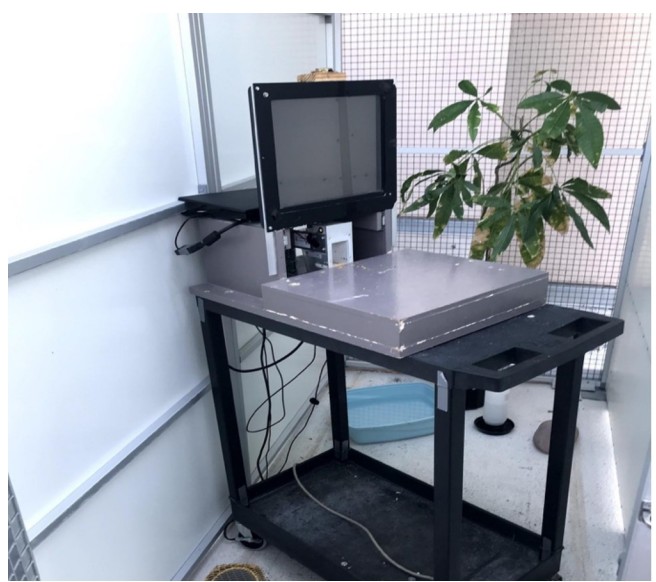
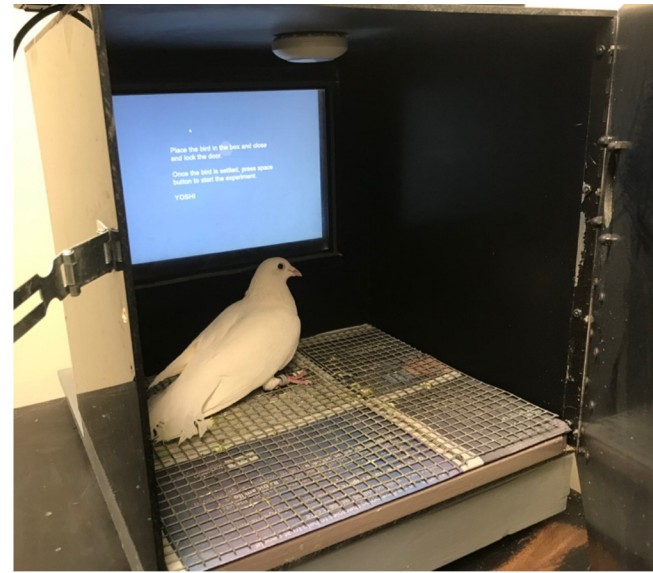

**Grackle TOC Setup**                                                      **Pigeon TOC Setup**

**Fig 1. Touchscreen-equipped operant chamber basic set up during food hopper training.** The computer monitor sits behind a touchscreen panel, and food is delivered via a food hopper that is moved by a robotic arm. The programs are run on a laptop located at the back of the touchscreen, and the laptop can be remotely controlled from outside the aviary using the experimenter's laptop. For the Grackles the TOC is not an enclosed chamber but for pigeons it is.

### The touchscreen-equipped operant chamber training procedure

Grackles were trained to interact with the TOC through three distinct steps: habituation to the TOC, hopper training, and training to peck the screen (see Table 2). The PsychoPy code for the below procedures (as well as the TOC experiments) is available at https://github.com/corinalogan/grackles/tree/master/Files/TouchscreenPsychoPy2code.

**1. Habituation to TOC apparatus.** Grackles were first habituated to the presence of the TOC. The TOC was placed in each aviary overnight with maintenance diet only accessible from the hopper and atop the platform in front of the hopper (i.e., food is only available on or in the TOC and no food is available from their regular food dish away from the TOC). Grackles that did not receive the TOC in their aviaries overnight were either birds that the experimenters thought might destroy the apparatus when left alone with it (e.g., Adobo) or those that participated readily in daytime sessions without prior overnight habituation (e.g., Yuca and Chilaquile). We actively encouraged the birds to approach the TOC monitor and eat Goldfish crackers out of the raised hopper during daytime sessions by sprinkling reward food (i.e., crackers) on the TOC platform in front of the monitor and hopper. Once grackles were comfortable eating from the raised hopper, grackles progressed to hopper training (below) to habituate the them to the sound and movement of the hopper and to learn they have a limited time period in which to access the food when the hopper is raised.

**Table 2. Outline of the three consecutive steps involved in training grackles to use a TOC.**

| 1. Habituation to TOC apparatus | 2. Hopper training | 3. Touch the screen for food |
|---|---|---|
| TOC present, but not turned on | TOC present, food hopper is active, touchscreen is turned off | TOC present and food hopper and touchscreen are on |

**2. Hopper training.** Grackles were trained to associate the sound and movement of the hopper being raised with the availability of food. To do so, all food was removed from the platform such that food (e.g., crushed up Goldfish crackers) was only available from within the hopper, and the experimenter began moving the hopper to deliver food to the birds when they approached it. This was made possible by remotely controlling the computer (*a DELL Inspiron 15*) attached to the TOC using a separate computer (*experimenter's computer*) outside of the testing area (Fig 1). Specifically, a free trial version of TeamViewer software (version 15.2.2756; www.teamviewer.com) was used to remotely control and run all training and testing programs on the TOC computer. A PsychoPy program was created to move the food hopper up or down with every press of the space bar on the experimenter's computer ("hopper training" program; file name: 1.Press_Space_for_food_2.Basic_mag_training_nostartscreen.psyexp). Grackles were habituated to the sound and movement of the hopper by first lowering the hopper remotely after an individual had eaten from the raised hopper. Once an individual was habituated to the sound and movement of the hopper lowering after reward delivery, the experimenter switched to raising the hopper remotely as the bird approached the hopper. When the bird learned that the noise of the hopper moving indicated that food was available, and they consistently approached and ate from the raised hopper, they progressed to the next stage of training in which they learned to touch the screen for the food reward (below).

**3. Touch the screen to obtain food.** Grackles were then trained to touch the screen to gain access to a food reward. A program was created that presented a small (2 cm x 2 cm) digital white square on the screen ("white square training" program, file name: 4.Food_Key_Only_2FullControl.psyexp). Pecks made directly to this stimulus resulted in an automatic 5 s of food access from the raised hopper, or the experimenter could press the spacebar on their laptop to raise the hopper manually. The initial intention was to use a mixed Pavlovian and Instrumental autoshaping procedure to encourage birds to engage with the TOC to obtain food rewards (i.e., the birds could peck a stimulus to receive the reward [Instrumental] and a trial always ended in reward regardless of their behavior [Pavlovian/Autoshaping]), but it became immediately clear after starting training that this would not work with the grackles who were not interested in interacting with the touchscreen unless they were hand-shaped to do so. This is in stark contrast to pigeons who readily engage in autoshaping procedures [35]. Instead, experimenters employed more basic hand-shaping procedures to encourage grackles to engage with the screen (i.e., the experimenter had to be present outside of the aviary to trigger the hopper to reward the bird for incrementally correct behaviors that eventually led to their touching the digital white square). Grackles were rewarded (by the experimenter cueing the hopper to move to the available position so the bird could eat) for first putting their bill on the screen near the digital white square, then closer to the square, until they were touching the square on their own and able to trigger the hopper themselves. However, the hand-shaping technique of remotely triggering the food hopper to get the grackle to peck the digital white square was not enough to get them to successfully engage with the digital white square. All subjects required additional hand-shaping methods (see S1 File for detailed descriptions of these methods).

## General touchscreen-equipped operant chamber training rules

Throughout the training process and during testing, it was crucial to make the TOC apparatus available to the bird only when the apparatus was ready for a bird to participate in a training trial. During daily testing and training, if the bird did not interact with the TOC within 5 minutes of it being placed inside their aviary, the TOC was removed and another session was attempted later. The aim was to train the birds that if they wanted to participate, they must do

so right away. Additionally, when they did choose to participate, we would stop the session and remove the TOC if they became distracted by interacting with different parts of the TOC or other objects in their aviary, or lost motivation and did not return to the TOC within 5 minutes. This likely resulted in the birds learning that if the TOC is available to them, they must stay engaged with it until completing the test. Consequently, testing sessions were likely able to be faster and more concentrated than if experimenters had waited for birds to participate.

## Results

### Habituation to TOC apparatus training

Habituation was conducted for between 0 to 12 nights, with the touchscreen being left in the aviary for additional habituation if grackles continued to avoid approaching it during daytime active training sessions. In previous experiments conducted by our group using experimentally naive pigeons, habituation to the TOC usually lasts only 2 days, but note that the pigeon TOCs are enclosed and birds are locked inside for approximately 30 minutes at a time (Blaisdell and Seitz, pers. obs.).

### Hopper training

It took grackles 3 to 25 days to pass hopper training ($n = 11$, Table 3, see S2 in S1 File for criterion). Hopper training was attempted with an additional three grackles, but two had to be released back to the wild before their training was complete, and one did not pass hopper training (Table 3). Experimentally naive pigeons typically exhibit reliable responding to the raised hopper after 2–3 days (1 session per day; Blaisdell and Seitz, pers. obs.).

### Touch the screen to obtain food

A total of 11 grackles received white square training and 10 of them passed (one had to be released back to the wild before his training was complete; Table 4, see S2 in S1 File for

**Table 3. Hopper training summary data.**

| Bird ID | Passed training | Number of days | Number of sessions | Number of trials |
|---|---|---|---|---|
| Fajita | No | 2 | 3 | 16 |
| Queso | Yes | 6 | 3+ | 25+ |
| Mole | Yes | 25 | 5+ | 21+ |
| Tomatillo | Yes | 5 | 4+ | 20+ |
| Habanero | Yes | 4 | 4+ | Not recorded |
| Chalupa | No | 5 | 6+ | 9+ |
| Tapa | Yes | 2 | 2 | 22 |
| Taco | Yes | 10 | 14 | 38 |
| Adobo | Yes | 4 | 6 | 81 |
| Diablo | Yes | 16 | 22 | 44+ |
| Burrito | Yes | 3 | 5 | 20+ |
| Marisco | No | 54 | 67 | 98 |
| Chilaquile | Yes | 3 | 4 | 40+ |
| Yuca | Yes | 6 | 12 | 56 |

The number of days is the total number of days included in the period from the first to the last training day (i.e., it is not the number of days on which training occurred), the number of sessions and trials include those where the bird did not participate. Note that when a number has a +, this means that the detailed data for the initial trials/sessions were not recorded. This is because we did not realize right away that collecting this data at the trial level would be useful for understanding how to better train grackles on the apparatus.

**Table 4. White square training summary data.**

| Bird ID | Passed training | Number of days | Number of sessions | Number of trials |
|---|---|---|---|---|
| Queso | Yes | 58+ | 22+ | 23+ |
| Mole | Yes | 12 | 7 | 80+ |
| Tomatillo | Yes | 3 | 12 | 91+ |
| Habanero | Yes | 43 | 49 | 204+ |
| Tapa | Yes | 3 | 5 | 53+ |
| Taco | No | 6 | 5 | 36 |
| Adobo | Yes | 7 | 13 | 98+ |
| Diablo | Yes | 8 | 17 | 126 |
| Burrito | Yes | 2 | 3 | 40+ |
| Chilaquile | Yes | 3 | 5 | 90+ |
| Yuca | Yes | 6 | 13 | 125+ |

The number of days is the total number of days included in the period from the first to the last training day (i.e., it is not only the number of days on which training occurred), the number of sessions and trials include those where the bird did not participate. Note that when a number has a +, this means that the detailed data for the initial trials and/or sessions were not recorded. This is because we did not realize right away that collecting this data at the trial level would be useful for understanding how to better train grackles on the apparatus.

criterion). It took 2 to 43 days to complete this training. Experimentally naive pigeons typically demonstrate high levels of responding to the screen after no more than 3 days of similar training (1 session per day; Blaisdell and Seitz, pers. obs.).

## Discussion

### Questions our limited training data can begin to answer (*post hoc*)

The data collected during TOC training allowed the beginning of a qualitative investigation a few *post hoc* insights. Such insights might be useful for programs that already run TOC experiments to determine whether there are ways to improve efficiency (e.g., what leads to faster training times) and how to decide which individuals to select to participate in these experiments. These insights could also help a researcher determine whether it would be feasible to implement a TOC experiment given the amount of time that is necessary for training.

To answer the below questions, the data from the procedures described above were examined for those birds who completed enough of the training (hopper training plus white square training) to determine whether they were a fast (<13 days to pass training) or slow learner (13 + days to pass training) (n = 11 grackles). The 13 day threshold was determined after comparing grackle training durations and was chosen to ensure that there were birds in both categories (fast and slow).

**Question 1:** Were those grackles that completed more experiments prior to beginning TOC training more likely to **complete TOC training faster**? If the answer to this question (or question 2) is yes, this is potentially because more experience with participating in any kind of experiment in general leads to the improvement of performance on any given future experiment (e.g.,[36]), and/or these individuals become more habituated to the aviary testing environment. In these cases, experimenters should put the TOC tests at the end of a test battery that involves both TOC and traditional procedures, or test individuals who already have extensive prior testing experience. **Answer 1:** The number of previously completed experiments (three or four) was likely not related to TOC training duration (Fig 2). All grackles, except Marisco, completed the maximum number of non-TOC tests possible before beginning TOC training and their training durations varied, sometimes substantially. Thus, participating in

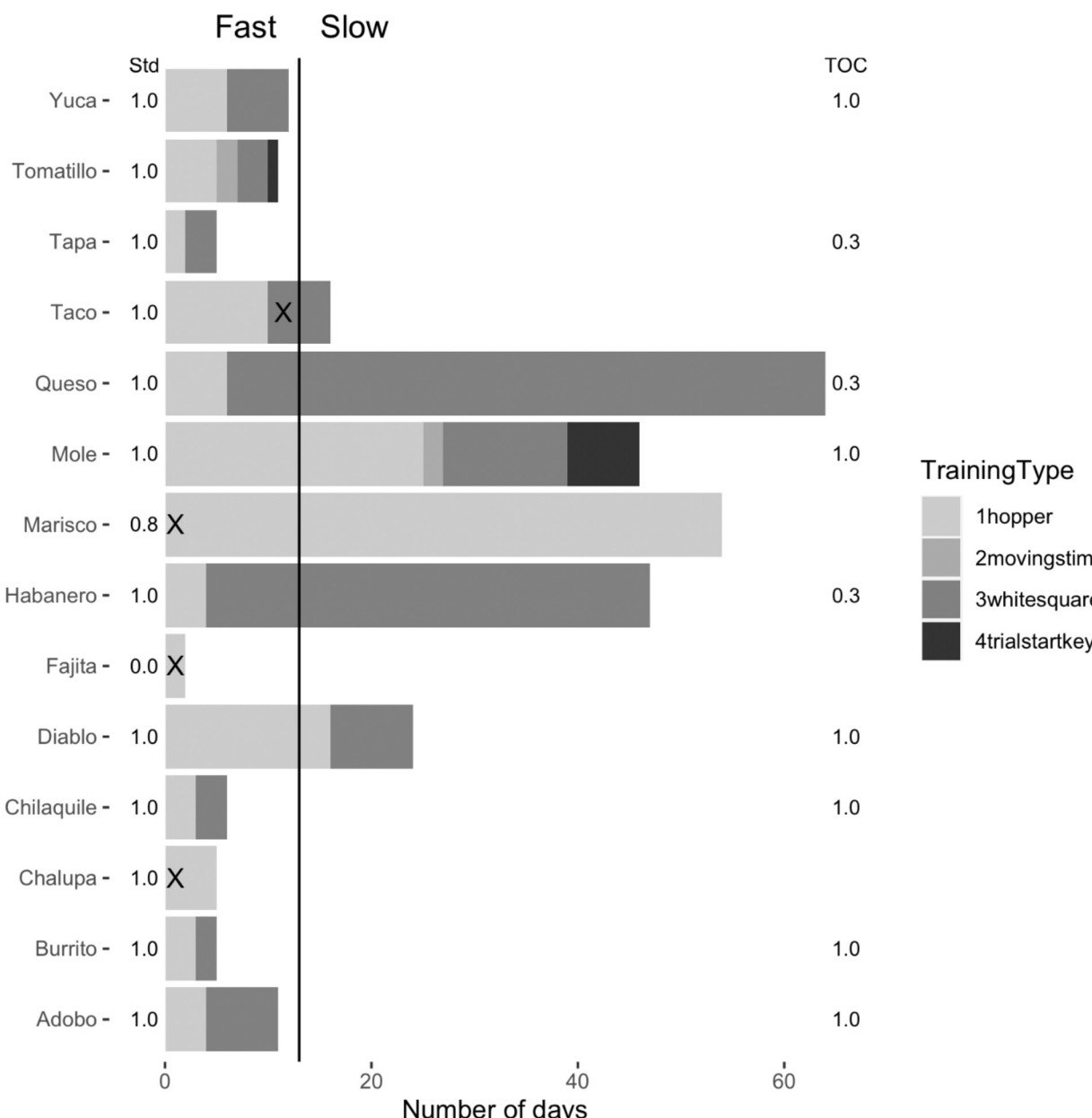

**Fig 2. Summary of the test battery order and performance, and TOC training duration (x axis) according to training type (hopper, moving stimulus, white square, and trial start key, always in that order) for each bird (y axis).** An "X" inside a bar of the bar plot indicates the bird did not pass that training type and their training ended there. The vertical line at 13 days indicates the threshold for fast (completed all TOC training <13 days) and slow (13+ days) learners. The proportion of non-TOC experiments that had been completed or were started by the time TOC training began is denoted by "Std", and the proportion of TOC experiments participated in after training completed is indicated by "TOC".

more experiments before beginning TOC training did not appear to result in faster, and thus more efficient, training.

**Question 2:** Were those grackles that completed more experiments prior to beginning TOC training more likely to **participate in more TOC tests**? **Answer 2:** The number of previously completed experiments (three or four) was likely not related to the number of TOC experiments they completed (Fig 2). The three grackles (Queso, Habanero, and Tapa) who did not complete all three TOC experiments did complete the maximum number of non-TOC

experiments before beginning TOC training. The exception, again, was Marisco who completed only three of the four experiments attempted and did not participate in any TOC experiments because he never passed TOC training. Thus, the amount of prior experimental participation by individuals seemed unrelated to their ability to complete TOC experiments (i.e., feasibility of conducting TOC experiments given individual competence in prior experiments).

**Question 3:** Were those grackles that completed TOC training faster more likely to **participate in more TOC experiments**? The amount of training time might be predicted to inversely relate to the number of TOC experiments they complete, potentially because those individuals who require less training time (i.e., are faster to reach criterion) may be more motivated to interact with and learn about the TOC. If such a relationship exists, this would be useful information for researchers who use TOCs because it would help them determine which individuals are more likely to complete testing, and thus which individuals to focus testing efforts on, if the research program is under time constraints. **Answer 3:** The duration of TOC training seemed unrelated to the number of TOC experiments grackles completed (Fig 2). For example, Mole and Habanero took a long time to pass TOC training, but Mole completed all three TOC experiments, while Habanero completed only one. In contrast, Tapa was a relatively fast TOC learner, but only completed one out of three TOC experiments.

In summary, the number of previously completed tests does not appear to indicate the likely TOC training speed or the likelihood of completing more TOC experiments, and TOC training speed did not appear to indicate how many TOC experiments they might complete. Although shortcuts for predicting grackle participation in TOC experiments were not found, these insights are offered in the hopes that this information will help direct future efforts in productive ways.

## Programming TOC experiments using PsychoPy programming software

One of the largest barriers that hinders researchers interested in animal cognition from conducting experiments on TOCs is that TOCs require tasks that must be written in some sort of computer programming language. There are a host of different options, many of which are not entirely user-friendly or intuitive for those without programming experience. Languages and programs that market themselves as more user-friendly, often require the purchase of costly licenses (e.g., E-Prime and MATLAB). To program tasks for the experiments investigating behavior in grackles (https://github.com/corinalogan/grackles/blob/master/README.md), and to study similar principles in pigeons and humans (https://pigeonrat.psych.ucla.edu/), we have had great success using PsychoPy and recommend this software for those interested in building their own TOCs to study animal cognition and beyond.

PsychoPy is a free and open-source application for programming experiments in the Python language and is compatible with most major operating systems (e.g., Windows, OS X, Linux) [37–39]. Programs can be written directly in Python, or in the 'Builder' view, a graphical user interface (GUI) which allows for a simpler production of a wide variety of stimuli. The builder mode makes generating stimuli and documenting appropriate responses simple and also allows for "Snippets" of code to be added into the "Routine" which allows for tremendous flexibility in terms of what can be presented and recorded during a task. There is also an active online support community (https://discourse.psychopy.org/) where hundreds of questions have already been addressed and new questions are swiftly attended to. Additionally, the latest version of PsychoPy (version 3 at the time of this publication) allows studies to be conducted online, which could be useful for studies where data are collected in different locations, for automatic online backup of the programs and data, and for easy sharing of programs among

researchers [40]. There are additional advantages for researchers interested in non-human versus human comparisons on identical or slightly modified tasks. While it was outside the scope of this article to fully explain the capabilities of PsychoPy, we highly recommend that those interested should consult a recently published textbook [41], which details how to use the program for creating experiments.

Instead, we will focus on how to use PsychoPy to create tasks that involve the operations of external hardware to deliver outcomes (e.g., food delivery via the food hopper in our grackle TOC) which is typical of a TOC setup. We then provide basic templates and advice on creating programs specifically for studying animal cognition—this should prevent some researchers from having to "reinvent the wheel." For more information and examples of our code, including a sample Go/No-Go inhibition task, please consult the following page (https://github.com/corinalogan/grackles/tree/master/Files/TouchscreenPsychoPy2code).

## Setup: Connecting the TOC to a food hopper

The first and perhaps most complicated step to setting up a TOC is establishing a connection between the program and an external hardware system that delivers some sort of outcome to the animal (e.g., food or shock). In the tasks described earlier, we used access to a food hopper used as reinforcement. The food hopper was attached to a robotic arm (Pololu Maestro 12 Channel USB Servo, Pololu Robotics, Las Vegas, NV) that could be raised or lowered. Thus, in the very beginning of the program for a given experiment, a Snippet of code was inserted to establish a connection between the program and the hopper. Given the universality of Python as a programming language, there may already exist online resources that explain how to control the external hardware using Python. That was the case for the Pololu system (https://github.com/FRC4564/Maestro). All experiments we conducted using this setup began with a brief 3 s Routine that established a connection between the program and the hardware and then another 3 s Routine that ensured the hopper was in the resting position out of reach from the subject (see Fig 3 for a screenshot of the experimental arrangement in PsychoPy).

**Basic programming outline.** Designing a Routine in PsychoPy builder mode is similar to creating a film in a video editing software in that it displays the temporal sequence of stimuli and responses that can occur throughout a given trial (see Fig 3). Different stimuli can be presented during each Routine and a number of different responses can be made. Additionally, the same Routine can be used to create an infinite number of different trials because the Routine can be conditionalized to an Excel file which determines which stimuli and responses will be presented (for more resources consult [39, 41]). We created a heavily annotated Go/No-Go sample program (https://github.com/corinalogan/grackles/tree/master/Files/TouchscreenPsychoPy2code/GoNoGo/2020-03AnnotatedGoNoGoCode) to illustrate these features. In that program, we created a Routine whereby a discriminative stimulus was presented above a food key, and the correct decision to peck or not peck the food key led to a reward. The features (e.g., color) of the discriminative stimulus and whether or not it would be rewarded was entirely contingent on the elements within the Excel file (see Figs 4 and 5).

Thus, for the majority of our experiments we created two main Routines, one that included all of the desired features of a trial and the other that controlled the hopper and resulted in a food reward or non-reward depending on the response that was made. These two Routines were enclosed in a loop that allowed the experimenter to determine the exact number of trials the animal would receive and which stimulus they would experience during those trials. Fig 3 provides an example of what this looked like in builder view, and an example of the code is provided in the Supplementary Material (S6 in S1 File). In the described sample Go/No-Go inhibition task, on some trials, a response to the food key terminated the Routine and led to

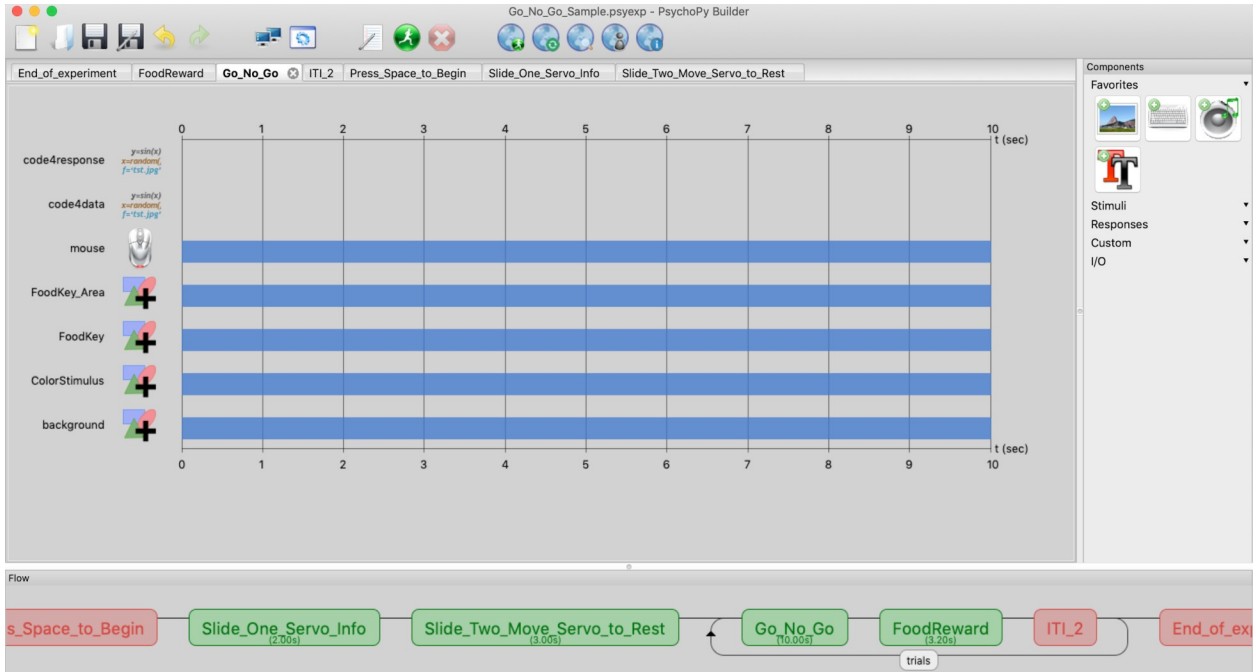

**Fig 3. Basic experimental setup in PsychoPy.** The "flow" (bottom bar) of the experiment represents the basic structure. The Press_Space_to_Begin "Routine" allows time for the experimenter to initiate the experiment on command. The first two Routines (Slide_One_Servo_Info and Slide_Two_Move_Servo_to_Rest) include code to establish a connection with the food hopper and then move it into the rest position. The rest of the experiment is generated within the loop titled, 'trials' (bottom bar).

the next Routine which was the food reward. Although PsychoPy is primarily used in an entirely graphical mode, extra capabilities can be achieved by specifying short snippets of custom code, written in the Python language, that run at specific times (such as at the beginning or end of the Routine, or on every screen refresh). For example, if a response was made to the food key while the non-reinforced stimulus was presented, the trial ended and a new argument was created that we coded as "skip_food_reward = True". The following food reward Routine contained a statement to immediately terminate the trial and not offer a food reward if skip_food_reward = True (see S7 in S1 File). In other words, the pecking response on a No-Go trial initiated a clause that skipped the food reward Routine that normally followed a trial. This resulted in no delivery of reinforcement as a result of the incorrect response. These simple snippets of code are intuitive even to those with little programming experience and are made easier to implement using the GUI.

|   | A | B | C | D |
|---|---|---|---|---|
| 1 | Trial | color | rewarded | iti_time |
| 2 | a | Hotpink | yes | 3 |
| 3 | b | green | no | 3 |

**Fig 4. Setting up the conditions in an Excel file that is attached to the PsychoPy program as it runs.** Psychopy will select a row from the attached excel file, and use the elements within that row to generate the trial. Thus, if row 2 were selected, the discriminative stimulus would be hot pink, a peck to the food key would be rewarded, and the inter-trial interval would be 3 seconds.

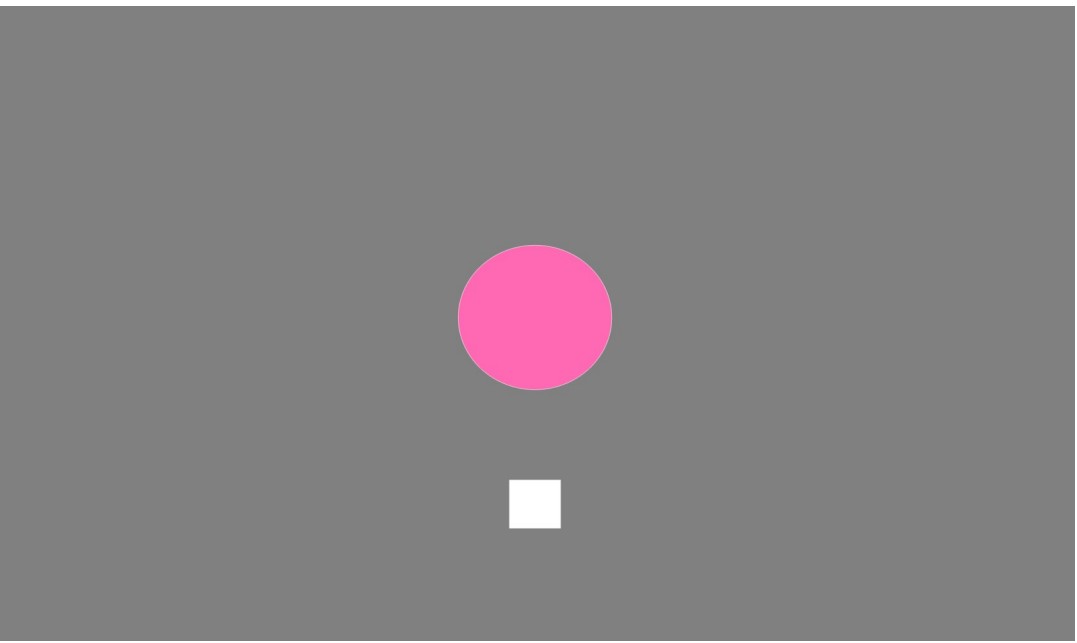

**Fig 5. Example trial in PsychoPy.** The color of the discriminative stimulus (circle) is dictated by the row chosen in the excel file, as is what happens when the food key is pecked.

## Food reward proxies for external programming

One of the most useful techniques involved simple stimuli provided by PsychoPy as proxies or representations for the movement of our external hardware (i.e., food hopper). Whenever we wanted to represent when the food hopper would be in the feeding position, we had a textbox appear on screen that said, "feeder is open". When we wanted to represent the feeder being lowered to the resting position, a different text box read, "feeder is closing". We then added a Snippet of code was then inserted that in essence posited: if the "*feeder is open"* textbox is on the screen, move the servo to the feeding position and if the "*feeder is closing"* textbox is on, move the feeder to rest. Thus, changing the duration of the food reward was as simple as changing the duration of the "feeder is open" textbox. We could also temporarily deactivate the code for the food hopper by adding a "#" in front of the argument (see S8 in S1 File). This allowed us to code experiments from any personal laptop (despite these computers not being connected to the hopper) anywhere in the world. Disconnecting the hopper also allowed the remote coder to test the program on the TOC without engaging the hopper. We could easily share programs among team members until the final program was agreed upon, then load it onto the TOC with the #'s removed and the text made invisible (by setting text font size to 0).

## Drawbacks of PsychoPy in studies of animal cognition

One noticeable drawback to using PsychoPy to operate TOCs and study animal cognition is that each TOC must be operated by an individual computer. That is, while some commonly used software (e.g., MedPC by Med Associates, Inc.) allows for a single computer to operate multiple operant chambers, our setup requires that each screen is operated by its own computer. This may pose some practicality issues for those interested in running squads of animals on the same procedure, however it is less problematic for running only a handful of animals at a time. With that said, we are not aware of any technical reasons as to why PsychoPy cannot be

used to control multiple monitors and external reinforcement machinery—but we are also not aware of any group who has successfully managed to construct such a setup.

## Concluding remarks

Touchscreen-equipped operant chambers confer a number of advantages over typical operant chambers and traditional laboratory-based techniques. Despite the reviewed challenges, we expect TOCs to continue to grow in popularity in a number of scientific disciplines. As some traditional laboratory procedures appear to easily transfer to a TOC, while others do not, future research should explore the causes of these discrepancies. We recommend that future studies continue to compare TOC experiments with standard tests to validate whether they measure the same ability. If comparisons show that TOC experiments are similar to the standard test, we suggest moving forward with the TOC test if a high level of automation without technical errors can be achieved. Additionally, it will be important to consider the amount of time using a TOC adds to the overall experiment duration. While it might often take longer to habituate and train individuals to use a TOC than a standard test, it might be worth it if it shortens the testing time and eliminates the need for an experimenter to be present.

After several years of training wild-caught animals to operate TOCs, we have used this article to share the advantages and disadvantages of our various approaches, report data from our ongoing experiments, and provide advice and programming suggestions for researchers interested in similar pursuits. The programs used throughout this project (and data) are accessible on open-science platforms, and the move from traditional OCs and open-field procedures to TOCs represents another opportunity to make behavioral science more precise, transparent, and replicable.

## Supporting information

**S1 File.**
(PDF)

**S1 Data.**
(CSV)

**S2 Data.**
(CSV)

## Acknowledgments

We thank Al Kamil and Debbie Kelly for brainstorming sessions on how to deal with the variety of problems we faced during training; Richard McElreath for project support; and our research assistants who helped trap the grackles and bring them into the aviaries: Aelin Mayer, Nancy Rodriguez, Brianna Thomas, Aldora Messinger, Elysia Mamola, Michael Guillen, Rita Barakat, Adriana Boderash, Olateju Ojekunle, August Sevchik, Justin Huynh, Jennifer Berens, Amanda Overholt, Michael Pickett, Mina Mohammed, Emily Blackwell, Kaylee Delcid, Brynna Hood, Samantha Bowser, Elise Lange, Sierra Planck, and Samuel Muñoz. For advice on using PsychoPy in this manuscript and technical support throughout this project, we thank Jon Peirce and Michael MacAskill.

## Author Contributions

**Conceptualization:** Benjamin M. Seitz, Kelsey McCune, Maggie MacPherson, Aaron P. Blaisdell, Corina J. Logan.

**Data curation:** Kelsey McCune, Maggie MacPherson, Luisa Bergeron, Corina J. Logan.

**Formal analysis:** Benjamin M. Seitz, Aaron P. Blaisdell, Corina J. Logan.

**Funding acquisition:** Corina J. Logan.

**Investigation:** Kelsey McCune, Maggie MacPherson, Luisa Bergeron, Corina J. Logan.

**Methodology:** Benjamin M. Seitz, Kelsey McCune, Luisa Bergeron, Aaron P. Blaisdell, Corina J. Logan.

**Project administration:** Kelsey McCune, Maggie MacPherson, Corina J. Logan.

**Resources:** Corina J. Logan.

**Software:** Benjamin M. Seitz.

**Supervision:** Aaron P. Blaisdell, Corina J. Logan.

**Visualization:** Benjamin M. Seitz, Kelsey McCune, Corina J. Logan.

**Writing – original draft:** Benjamin M. Seitz, Kelsey McCune, Maggie MacPherson, Luisa Bergeron, Corina J. Logan.

**Writing – review & editing:** Benjamin M. Seitz, Kelsey McCune, Maggie MacPherson, Luisa Bergeron, Aaron P. Blaisdell, Corina J. Logan.

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
