## [Decision Letter · Decision Letter 0]

16 Nov 2020

PONE-D-20-31716

Using Touchscreen Equipped Operant Chambers to Study Comparative Cognition. Benefits, Limitations, and Advice

PLOS ONE

Dear Dr. Seitz,

Thank you for submitting your manuscript to PLOS ONE. We have now received two reviews for the manuscript. As you will notice, both of them see merit in the paper, but have a number of constructive suggestions to further enhance the manuscript. We invite you to submit a revised version of the manuscript that addresses the points raised during the review process.

We look forward to receiving your revised manuscript.

Kind regards,

Rajagopalan Srinivasan

Academic Editor

PLOS ONE

Journal Requirements:

2.) Your ethics statement should only appear in the Methods section of your manuscript. If your ethics statement is written in any section besides the Methods, please move it to the Methods section and delete it from any other section. Please ensure that your ethics statement is included in your manuscript, as the ethics statement entered into the online submission form will not be published alongside your manuscript.

3.)Thank you for stating the following in the Acknowledgments Section of your manuscript:

[Benjamin Seitz

is supported by National Science Foundation grant DGE-1650604. Aaron P. Blaisdell is

supported by National Science Foundation research grant BCS-1844144.]

 [This research is funded by the Department of Human Behavior, Ecology and Culture at the Max Planck Institute for Evolutionary Anthropology (https://www.eva.mpg.de/index.html) (2017-current), and by a Leverhulme Early Career Research Fellowship (https://www.leverhulme.ac.uk/early-career-fellowships) to CJL (2017-2018). The funders had no role in study design, data collection and analysis, decision to publish, or preparation of the manuscript.]

4.) Please note that in order to use the direct billing option the corresponding author must be affiliated with the chosen institute. Please either amend your manuscript to change the affiliation or corresponding author, or email us at plosone@plos.org with a request to remove this option.

Reviewers' comments:

Reviewer's Responses to Questions

**Comments to the Author**

1. Is the manuscript technically sound, and do the data support the conclusions?

Reviewer #1: Yes

Reviewer #2: Yes

2. Has the statistical analysis been performed appropriately and rigorously? 

Reviewer #1: N/A

Reviewer #2: N/A

3. Have the authors made all data underlying the findings in their manuscript fully available?

Reviewer #1: Yes

Reviewer #2: Yes

4. Is the manuscript presented in an intelligible fashion and written in standard English?

Reviewer #1: Yes

Reviewer #2: Yes

5. Review Comments to the Author

Reviewer #1: Review of: Using Touchscreen Equipped Operant Chambers to Study Comparative Cognition. Benefits, Limitations, and Advice.

By Seitz et al.

This manuscript offers a unique perspective regarding the use of wild-caught birds as experimental subjects and, as well, on the use of touchscreen-based operant chambers to study cognition. The authors offer various resources and information on how to use and set up touchscreen chambers and programs, which could prove valuable for researchers. I appreciate the authors’ emphasis on making this information and methodology easily accessible and open source. I believe this manuscript could be a valuable contribution to the literature, following considerations of the points below.

Major points:

Though the manuscript is submitted under the category of 'Research Article', it reads more as a handbook or case report. Moreover, the language used is very often too informal and reads like a casual conference talk. This makes the manuscript hard to follow. The authors did conduct experiments that yielded data, so the manuscript could easily be shaped into a traditional research article while still maintaining that some aspects of TOC/PsychoPy set-up and usage. I suspect readers will find the experiments just as (if not more) useful and interesting, especially regarding how to train animals with TOCs. Moreover, the title mentions "comparative cognition", though the current manuscript is lacking justification regarding how cognition is measured or in describing the types of tasks that TOC offers that better study cognition in relation to human subjects (or other species). Restructuring the manuscript to describe the specifics of the tasks employed would aid clarity.

As in standard research articles, the Methods and Procedures sections would address the following points that are currently missing:

Describe the training protocol (e.g., non-TOC training) and criteria for subjects to move onto the next phase.

Specifics of each TOC program (stimuli presentation frequency and duration, intertrial intervals, response requirement, consequence of correct/incorrect responses). It appears a go/no-go task was used and a simple discrimination task – what are these program parameters? What stimuli were used?

Measurement of behavioral performance (Accuracy? Reaction time? Number of trials/sessions completed? Typical session length? Number of opportunities to engage with the touchscreen? Etc).

Similarly, the manuscript needs a dedicated 'Subjects' section that describes how the birds were caught, handled, housed, and fed – especially since one of the authors' main points is offering grackles as an option for experimental subjects. Were they food-restricted? Did the birds live inside these TOCs or were they housed in a vivarium? This section could also include instances where birds needed medical procedures or needed to be excluded from certain studies.

Finally, and importantly, I recommend that the authors substantially reduce the amount of first-person language, which will make the manuscript considerably less informal.

Other suggestions:

While advantages and disadvantages of touchscreens are appreciated, this section can be condensed into 2-3 paragraphs, as these points have been previously detailed elsewhere.

The training protocol as a google doc can be condensed and described in a Methods and Procedures section. Moreover, anyone with this link can edit and make changes to this document. If after considering these revisions and the authors still deem the google doc necessary, transfer that information into the supplementary materials.

First paragraph, p. 9: The authors mention an interesting point that testing was faster due to less interference from experimenters. Is there data to support this or previous literature to cite that might support this claim? Moreover, is this a feature that only touchscreens offer? What about non-touchscreen chambers that use an automated system in an enclosed chamber, but not necessarily touchscreen? Please clarify.

The last sentence of this paragraph suggests that TOCs might be able to "eliminate the need for these experiments to occur in captivity." This is, at best, overstated as many studies require laboratory-bred species and highly controlled environments (e.g., genetic, neural, pharmacological, toxicological, etc). The authors should consider removing this point.

p. 10: Authors stated: "This resulted in the birds learning that if they wanted to participate, they must do so right away." Is there data to support this? Showing how many birds went to the touchscreen immediately upon presentation and how many of those completed the session or task without interruption would be helpful for other researchers. Moreover, did the touchscreen live in the chamber and was only turned on when needed? Or was it wheeled in? These details should be described in a Methods/Procedures section.

p. 11: For ease of coherence and overall organization, this section on questions should be reserved for the Discussion section. Results should merely be reported. Moreover, to question 1 and question 3, the authors provide speculation that appears to be unfounded, based on the results of the current studies. For example: "the amount of training time might be predicted to inversely relate to the number of TOC experiments they complete, potentially because those individuals who require less training time may be more motivated to interact with and learn about the TOC..." to which these factors are actually "unrelated".

Last paragraph, p. 11: Use of the general term "test-taking" as in "test-taking experience" and "better test-takers" conflates performance to a specific task or set of tasks that may not be generalize to performance across all types of "tests". I think the authors are trying to state that experience with interacting with touchscreen experience begets better performance on other types of touchscreen tasks learned subsequently.

Consider replacing Table 1 with a visual display of duration of training for the various phases. This table is very difficult to read and does not allow readers to see patterns (or the lack thereof) with non-TOC tests and subsequent TOC outcomes, for example. The same critique applies to Table 2 and both could possibly be displayed in the same graph.

It is not immediately clear what this distinction between fast and slow learners describe. Does it refer to completion of all experiments (non-TOC and TOC)? Or just TOC experiments? Outlining these details in a methods section would provide clarity. And rather than designating subjects as fast or slow, reporting the number of days/sessions required to complete training would be more descriptive. Or even better would be to include these in a graph (see above comment).

I appreciate the use and descriptions of PsychoPy terms, and consistency should be exercised when referring to these events. For example, should 'routine' be capitalized? If not, italicize on first usage and leave uncapitalized (but check journal guidelines if available first). Moreover, snippet is used in quotes on p. 15 but was mentioned earlier on p. 14 in single quotes.

Reviewer #2: Reviewer Comments:

The manuscript from Seitz and colleagues investigates the use of touchscreen operant conditioning in the study of comparative cognition. The manuscript focuses on the implementation of the touchscreen operant platform for non-model species, specifically wild-caught great-tailed grackles. The authors provide useful insights into the training and assessment parameters of grackles in touchscreen operant conditioning. The authors also discuss modifications made to the hardware platform to facilitate animal engagement. It was also great to see the accompanying sample video of the training protocol. It would be nice to see other manuscripts doing the same.

That being said, there are some limitations that should be addressed. Specific comments are below;

Major Comments:

1. The manuscript focuses on the training of wild-caught great-tailed grackles and the comparison is made to traditional laboratory pigeons (Page 8, line 16). However, there is no direct comparison between training timelines (number of training days or trials) between grackles and laboratory pigeons. It would be important to include data from laboratory pigeons (for example, the number of training days/trials on similar tasks.) and discuss the differences/similarities.

2. The authors should discuss within-session task parameters that maybe important in assessing or predicting task performance, such as the number of screen approaches or stimulus response delay (i.e. the time taken from stimulus onset to screen touch).

Minor comments:

1. It would be useful if the authors included a figure which compared the touchscreen equipment used for pigeons and the modified equipment for the grackles.

2. The authors should include a brief comment in the discussion comparing the training time of TOC to standard behavioural tasks in grackles.

3. The authors should include a brief discussion on potential future cognitive test battery, should researchers only use TOC or a combination of TOC and standard tests?

4. One of the important benefits of TOC is the translational ability, that is the ability to perform human neuropsychiatric assessment in animal models. This should be mentioned in the “TOC advantages section” on page 4.

6. PLOS authors have the option to publish the peer review history of their article (what does this mean?). If published, this will include your full peer review and any attached files.

Reviewer #1: No

Reviewer #2: **Yes: **James O'Leary

---

## [Author Response · Author response to Decision Letter 0]

29 Dec 2020

Reviewer #1: 

Review of: Using Touchscreen Equipped Operant Chambers to Study Comparative Cognition. Benefits, Limitations, and Advice.

By Seitz et al.

COMMENT 1: This manuscript offers a unique perspective regarding the use of wild-caught birds as experimental subjects and, as well, on the use of touchscreen-based operant chambers to study cognition. The authors offer various resources and information on how to use and set up touchscreen chambers and programs, which could prove valuable for researchers. I appreciate the authors’ emphasis on making this information and methodology easily accessible and open source. I believe this manuscript could be a valuable contribution to the literature, following considerations of the points below.

***RESPONSE 1: Thank you very much for your feedback! We look forward to addressing your comments below.

Major points:

COMMENT 2: Though the manuscript is submitted under the category of 'Research Article', it reads more as a handbook or case report. Moreover, the language used is very often too informal and reads like a casual conference talk. This makes the manuscript hard to follow. The authors did conduct experiments that yielded data, so the manuscript could easily be shaped into a traditional research article while still maintaining that some aspects of TOC/PsychoPy set-up and usage. I suspect readers will find the experiments just as (if not more) useful and interesting, especially regarding how to train animals with TOCs. Moreover, the title mentions "comparative cognition", though the current manuscript is lacking justification regarding how cognition is measured or in describing the types of tasks that TOC offers that better study cognition in relation to human subjects (or other species). Restructuring the manuscript to describe the specifics of the tasks employed would aid clarity.

***RESPONSE 2: We changed the tone to be more formal and we structured it to follow a research article format by adding clear Methods, Results, and Discussion sections. See Response 3 for how we addressed incorporating information about the TOC experiments.

We have also changed the word “Comparative” in the title to “Animal.” This more clearly highlights the goal of our paper: to provide important information to those who wish to study animal cognition using TOCs. 

COMMENT 3: As in standard research articles, the Methods and Procedures sections would address the following points that are currently missing:

Describe the training protocol (e.g., non-TOC training) and criteria for subjects to move onto the next phase.

Specifics of each TOC program (stimuli presentation frequency and duration, intertrial intervals, response requirement, consequence of correct/incorrect responses). It appears a go/no-go task was used and a simple discrimination task – what are these program parameters? What stimuli were used?

Measurement of behavioral performance (Accuracy? Reaction time? Number of trials/sessions completed? Typical session length? Number of opportunities to engage with the touchscreen? Etc).

***RESPONSE 3: We moved the TOC training protocol, including the specifics of each TOC program, and passing criteria from the Supplementary Material to the new Methods section. In this article, we present only data from the TOC training procedures. The data and detailed methodology for the non-TOC experiments and for the TOC experiments that these individuals experienced are presented in other articles:

- TOC go/no go: Are the more flexible individuals also better at inhibition?

- TOC causal cognition: Do the more flexible individuals rely more on causal cognition? Observation versus intervention in causal inference in great-tailed grackles

- TOC reversal learning: Is behavioral flexibility manipulatable and, if so, does it improve flexibility and problem solving in a new context?

- Non-TOC reversal learning and puzzlebox: Is behavioral flexibility manipulatable and, if so, does it improve flexibility and problem solving in a new context?

- Non-TOC individual differences assays: Is behavioral flexibility linked with exploration, but not boldness, persistence, or motor diversity?

- Non-TOC demonstrator training for a social learning experiment: Investigating the use of learning mechanisms in a species that is rapidly expanding its geographic range

To clarify this, we added a section to the manuscript titled Test Battery:

“Before beginning TOC training to prepare the birds for the TOC experiments (go/no go: see Logan et al. 2019b for details; causal cognition: see Blaisdell et al. 2019; reversal learning: see Logan et al. 2019a), these individuals experienced non-TOC experiments. Non-TOC experiments included a color tube reversal learning experiment and a puzzle box experiment (see Logan et al. 2019a for details), a detour experiment (see Logan et al. 2019b), exploration and boldness assays (see McCune et al. 2019a), and demonstrator training for a social learning experiment (McCune et al. 2019c).”

We also added a table for this section, where we summarize which TOC and non-TOC experiments each bird participated in and the order in which the experiments were experienced.

COMMENT 4: Similarly, the manuscript needs a dedicated 'Subjects' section that describes how the birds were caught, handled, housed, and fed – especially since one of the authors' main points is offering grackles as an option for experimental subjects. Were they food-restricted? Did the birds live inside these TOCs or were they housed in a vivarium? This section could also include instances where birds needed medical procedures or needed to be excluded from certain studies.

***RESPONSE 4: We moved our Subjects section from Supplementary Materials to Methods. And we moved the description of Queso’s medical procedure and the explanation of the birds who did not complete all of the tests from the Table 1 legend to the Subjects section.

COMMENT 5: Finally, and importantly, I recommend that the authors substantially reduce the amount of first-person language, which will make the manuscript considerably less informal.

***RESPONSE 5: Where possible, we changed the language from first person to third person. In areas where we describe our own setup or interpret our own results we have kept first person language—a style many agree to be more succinct and scientific (Pinker, 2014).

Pinker, S. 2014. The Sense of Style: The Thinking Person’s Guide to Writing in the 21st Century. London: Allen Lane.

COMMENT 6: Other suggestions:

While advantages and disadvantages of touchscreens are appreciated, this section can be condensed into 2-3 paragraphs, as these points have been previously detailed elsewhere.

***RESPONSE 6: We have reduced this section substantially, going from 1671 words to 735 words and from 6 to 3 paragraphs.

COMMENT 7: The training protocol as a google doc can be condensed and described in a Methods and Procedures section. Moreover, anyone with this link can edit and make changes to this document. If after considering these revisions and the authors still deem the google doc necessary, transfer that information into the supplementary materials.

***RESPONSE 7: Thank you for pointing out that the google doc was editable by anyone with the link! We didn’t quite realize that and have now changed it so it is not editable. We would like to keep the training protocol in the fully detailed form because it will make it easier for others to replicate our methods, therefore, we added the training protocol as Supplementary Material 2.

COMMENT 8: First paragraph, p. 9: The authors mention an interesting point that testing was faster due to less interference from experimenters. Is there data to support this or previous literature to cite that might support this claim? Moreover, is this a feature that only touchscreens offer? What about non-touchscreen chambers that use an automated system in an enclosed chamber, but not necessarily touchscreen? Please clarify. 

***RESPONSE 8: The data for the timing of reversal learning sessions using a standard test and for a TOC apparatus will soon be available at Logan et al. (2019 http://corinalogan.com/Preregistrations/g_flexmanip.html). With the standard test, the limiting factor is how fast the experimenter can go in and reset each trial. We looked at the participation data in the standard test for one of our fastest birds, Queso, where we were able to conduct 25 trials in 21 minutes, 21 trials in 11 minutes, and 20 trials in 10 minutes in separate test sessions. In contrast, reversal trials on the TOC could be conducted in half the time: Queso completed 19 trials in 6.5 minutes in two separate sessions. We added the citation for the data to this paragraph so it is clear where we draw our conclusion from. The increased testing speed is not unique to touchscreens, and could be achieved by other automated systems that eliminate the need for an experimenter to enter the testing room between trials. We noted this in the revised version:

“(automation makes testing faster; see Logan et al. 2019a for data and Perone, 1991 for discussion on benefits of free operant techniques)”

COMMENT 9: The last sentence of this paragraph suggests that TOCs might be able to "eliminate the need for these experiments to occur in captivity." This is, at best, overstated as many studies require laboratory-bred species and highly controlled environments (e.g., genetic, neural, pharmacological, toxicological, etc). The authors should consider removing this point.

***RESPONSE 9: Sorry for the confusion! We actually meant to say that if it was possible to conduct TOC experiments in the wild, that would be preferable for many research programs investigating species in the wild because it would eliminate the extra time and costs of bringing individuals temporarily into captivity. We rephrased to:

“Additionally, if the apparatus and training can be effectively modified, it could facilitate bringing the TOC to the field, which could remove the additional time and financial costs of temporary captivity for research programs that focus on wild individuals”

COMMENT 10: p. 10: Authors stated: "This resulted in the birds learning that if they wanted to participate, they must do so right away." Is there data to support this? Showing how many birds went to the touchscreen immediately upon presentation and how many of those completed the session or task without interruption would be helpful for other researchers. Moreover, did the touchscreen live in the chamber and was only turned on when needed? Or was it wheeled in? These details should be described in a Methods/Procedures section.

***RESPONSE 10: This was our general testing policy based on Logan’s previous experience of testing birds. Because we didn’t conduct an experiment to determine whether this practice makes a difference in participation speed, there are no data to support this claim. Accordingly, we updated Methods > Touchscreen-equipped Operant Chamber Training Rules (changes in bold):

“Although we did not experimentally determine whether these training rules affected the bird’s behavior, we hoped this resulted in the birds learning that if the TOC is available to them, they must stay engaged with it until completing the test.”

Unfortunately, we do not have data on the latency to touch the screen after the start of a training trial. We often have data on whether a bird participated at all during a given testing session or whether it did not participate in even one training trial (see the data associated with our article where the Choice column=-1 when a bird did not participate in that trial or session). Details about when the touchscreen was on or off are in the Methods > Training procedure, which we moved up from the Supplementary Material, and we added to the Subjects section:

“Unless habituation was occurring, the TOC was wheeled into the aviary, a training session was conducted, and then it was wheeled out of the aviary again.”

COMMENT 11: p. 11: For ease of coherence and overall organization, this section on questions should be reserved for the Discussion section. Results should merely be reported. Moreover, to question 1 and question 3, the authors provide speculation that appears to be unfounded, based on the results of the current studies. For example: "the amount of training time might be predicted to inversely relate to the number of TOC experiments they complete, potentially because those individuals who require less training time may be more motivated to interact with and learn about the TOC..." to which these factors are actually "unrelated".

***RESPONSE 11: Good point that these post hoc questions should be moved to the discussion. We made this change. Thank you for pointing out this source of confusion. We revised this section to read (additions in bold): 

“The amount of training time might be predicted to inversely relate to the number of TOC experiments they complete, potentially because those individuals who require less training time (i.e., are faster to reach criterion) may be more motivated to interact with and learn about the TOC” 

COMMENT 12: Last paragraph, p. 11: Use of the general term "test-taking" as in "test-taking experience" and "better test-takers" conflates performance to a specific task or set of tasks that may not be generalize to performance across all types of "tests". I think the authors are trying to state that experience with interacting with touchscreen experience begets better performance on other types of touchscreen tasks learned subsequently.

RESPONSE 12: Thank you for bringing this up - we weren’t very clear with the way we phrased this. We revised the sentence to say:

Questions our limited training data can begin to answer (post hoc): “If the answer to this question (or question 2) is yes, this is potentially because more experience with participating in any kind of experiment in general leads to the improvement of performance on any given future experiment (e.g., Thornton & Lukas 2012), and/or these individuals become more habituated to the aviary testing environment.”

COMMENT 13: Consider replacing Table 1 with a visual display of duration of training for the various phases. This table is very difficult to read and does not allow readers to see patterns (or the lack thereof) with non-TOC tests and subsequent TOC outcomes, for example. The same critique applies to Table 2 and both could possibly be displayed in the same graph.

***RESPONSE 13: Good idea! We made a new figure that shows Tables 1 and 2 in a stacked barplot format, and we deleted Tables 1 and 2 while moving them into the data set at KNB.

COMMENT 14: It is not immediately clear what this distinction between fast and slow learners describe. Does it refer to completion of all experiments (non-TOC and TOC)? Or just TOC experiments? Outlining these details in a methods section would provide clarity. And rather than designating subjects as fast or slow, reporting the number of days/sessions required to complete training would be more descriptive. Or even better would be to include these in a graph (see above comment).

***RESPONSE 14: Sorry for the confusion! We see now that this detail was buried in a paragraph and could easily be missed. We now separated it out into its own paragraph, which appears just before the first question:

“To answer the below questions, the data in Table 1 were examined for those birds who completed enough of the training (hopper training plus white square training) to determine whether they were a fast (<13 days to pass training) or slow learner (13+ days to pass training) (n=11 grackles).”

We added to the data sheet that goes with the new figure 2 the number of days to pass training for each bird (located at the KNB data repository).

COMMENT 15: I appreciate the use and descriptions of PsychoPy terms, and consistency should be exercised when referring to these events. For example, should 'routine' be capitalized? If not, italicize on first usage and leave uncapitalized (but check journal guidelines if available first). Moreover, snippet is used in quotes on p. 15 but was mentioned earlier on p. 14 in single quotes.

***RESPONSE 15: We thank the reviewer for pointing out these inconsistencies. We have capitalized all mentions of Routines per a recent publication by the founders of PsychoPy and will work with copy editors if changes are necessary. 

https://link.springer.com/article/10.3758/s13428-018-01193-y

Reviewer #2: Reviewer Comments:

COMMENT 16: The manuscript from Seitz and colleagues investigates the use of touchscreen operant conditioning in the study of comparative cognition. The manuscript focuses on the implementation of the touchscreen operant platform for non-model species, specifically wild-caught great-tailed grackles. The authors provide useful insights into the training and assessment parameters of grackles in touchscreen operant conditioning. The authors also discuss modifications made to the hardware platform to facilitate animal engagement. It was also great to see the accompanying sample video of the training protocol. It would be nice to see other manuscripts doing the same.

That being said, there are some limitations that should be addressed. Specific comments are below;

***RESPONSE 16: Thank you for your positive assessment! We address your comments below.

COMMENT 17: Major Comments:

1. The manuscript focuses on the training of wild-caught great-tailed grackles and the comparison is made to traditional laboratory pigeons (Page 8, line 16). However, there is no direct comparison between training timelines (number of training days or trials) between grackles and laboratory pigeons. It would be important to include data from laboratory pigeons (for example, the number of training days/trials on similar tasks.) and discuss the differences/similarities.

***RESPONSE 17: This is a valid critique. Now in the results section, we compare the average number of days that grackles took to complete each major training procedure on a TOC to the performance of experimentally naive pigeons on those same tasks. Unfortunately, we do not have an abundance of hard data to support or help visualize the pigeon estimates. This is because we have never thought it necessary to keep track of how long it takes to train our pigeons to use our TOC systems--we typically just train until “they’re ready” which means pecking the screen at some asymptotic level. With that said, we have now trained over 50 pigeons to operate TOCs and are confident that the relative estimates are an accurate representation of how quickly this species learns to engage with TOCs. Specifically the following sentences have been added in the MS.

Page 15: “This is in stark contrast to pigeons who will readily engage in autoshaping procedures (Brown & Jenkins, 1968)”

Page 23: “In previous experiments conducted by our group using experimentally naive pigeons, habituation to the TOC usually lasts only 2 days, but note that the pigeon TOCs are enclosed and birds are locked inside for approximately 30 minutes at a time (Blaisdell and Seitz, pers. obs.).”

Page 25: “Experimentally naive pigeons typically exhibit reliable responding to the raised hopper after 2-3 days (1 session per day).”

Page 24: “Experimentally naive pigeons typically demonstrate high levels of responding to the screen after no more than 3 days of similar training (1 session per day).”

COMMENT 18: 2. The authors should discuss within-session task parameters that maybe important in assessing or predicting task performance, such as the number of screen approaches or stimulus response delay (i.e. the time taken from stimulus onset to screen touch).

***RESPONSE 18: We agree that there could be any number of useful variables to examine within a session that might be related to task performance. Unfortunately, we did not collect data on these variables because we originally thought that it would be much easier to train the grackles to use the TOC and that the methods would be similar to the methods used on pigeons. This meant that training sessions were only video recorded when the experimenter was attempting to get the bird to pass criterion, and so the bulk of the training sessions were not video recorded. We struggled through a variety of programs and tricks to figure out what would work for each bird, which meant that we did not have a priori hypotheses or analysis plans. It was only after struggling that we thought it would be a good idea to write an article about these experiences, to help others get more out of their training data and set them up to conduct experiments on how to decrease training times, and so on.

COMMENT 19: Minor comments:

1. It would be useful if the authors included a figure which compared the touchscreen equipment used for pigeons and the modified equipment for the grackles.

***RESPONSE 19: Thank you for this suggestion, we have now revised figure 1 to also include a photo of the pigeon setup.

COMMENT 20: 2. The authors should include a brief comment in the discussion comparing the training time of TOC to standard behavioural tasks in grackles.

***RESPONSE 20: Thanks for catching this! We added this to Concluding Remarks:

“Additionally, it will be important to consider the amount of time that using a TOC adds to the overall experiment duration. While it might often take longer to habituate and train individuals to use a TOC than a standard test, it might be worth it if it shortens the testing time and eliminates the need for an experimenter to be present.”

COMMENT 21: 3. The authors should include a brief discussion on potential future cognitive test battery, should researchers only use TOC or a combination of TOC and standard tests?

***RESPONSE 21: Great point! We added this to Concluding Remarks:

“We recommend that future studies continue to compare TOC experiments with standard tests to validate whether they measure the same ability. If comparisons show that TOC experiments are similar to the standard test, we suggest moving forward with the TOC test if a high level of automation without technical errors can be achieved.”

COMMENT 22: 4. One of the important benefits of TOC is the translational ability, that is the ability to perform human neuropsychiatric assessment in animal models. This should be mentioned in the “TOC advantages section” on page 4.

 ***RESPONSE 22: This is an excellent suggestion. This point now concludes our advantages of TOCs section. “Finally, because humans and non-humans can be given nearly identical behavioral tasks in terms of the stimuli presented and available responses (but not necessarily reinforcement received), TOCs can be powerful tools for translational research especially for studying neuropsychiatric phenomena (Bussey, 2012).”

6. PLOS authors have the option to publish the peer review history of their article (what does this mean?). If published, this will include your full peer review and any attached files.

Do you want your identity to be public for this peer review? For information about this choice, including consent withdrawal, please see our Privacy Policy.

Reviewer #1: No

Reviewer #2: Yes: James O'Leary

---

## [Decision Letter · Decision Letter 1]

20 Jan 2021

Using Touchscreen Equipped Operant Chambers to Study Animal Cognition. Benefits, Limitations, and Advice

PONE-D-20-31716R1

Dear Dr. Seitz,

We’re pleased to inform you that your manuscript has been judged scientifically suitable for publication and will be formally accepted for publication once it meets all outstanding technical requirements.

Kind regards,

Rajagopalan Srinivasan

Academic Editor

PLOS ONE

Additional Editor Comments (optional):

Reviewers' comments:

Reviewer's Responses to Questions

**Comments to the Author**

1. If the authors have adequately addressed your comments raised in a previous round of review and you feel that this manuscript is now acceptable for publication, you may indicate that here to bypass the “Comments to the Author” section, enter your conflict of interest statement in the “Confidential to Editor” section, and submit your "Accept" recommendation.

Reviewer #1: All comments have been addressed

Reviewer #2: All comments have been addressed

2. Is the manuscript technically sound, and do the data support the conclusions?

Reviewer #1: Yes

Reviewer #2: Yes

3. Has the statistical analysis been performed appropriately and rigorously? 

Reviewer #1: Yes

Reviewer #2: Yes

4. Have the authors made all data underlying the findings in their manuscript fully available?

Reviewer #1: Yes

Reviewer #2: Yes

5. Is the manuscript presented in an intelligible fashion and written in standard English?

Reviewer #1: Yes

Reviewer #2: Yes

6. Review Comments to the Author

Reviewer #1: The authors responded to my suggestions well and did a very nice job revising. I congratulate them on a fine contribution to the field.

Reviewer #2: The authors have addressed all previous comments and should be commended for their work. I would recommend the manuscript for publication.

7. PLOS authors have the option to publish the peer review history of their article (what does this mean?). If published, this will include your full peer review and any attached files.

Reviewer #1: No

Reviewer #2: **Yes: **James O'Leary

---

## [Editor Report · Acceptance letter]

28 Jan 2021

PONE-D-20-31716R1 

Using Touchscreen Equipped Operant Chambers to Study Animal Cognition. Benefits, Limitations, and Advice. 

Dear Dr. Seitz:

I'm pleased to inform you that your manuscript has been deemed suitable for publication in PLOS ONE. Congratulations! Your manuscript is now with our production department. 

Kind regards, 

on behalf of

Dr. Rajagopalan Srinivasan 

Academic Editor

PLOS ONE